# Investigation of Antibacterial Properties of Corrosion-Resistant 316L Steel Alloyed with 0.2 wt.% and 0.5 wt.% Ag

**DOI:** 10.3390/ma16010319

**Published:** 2022-12-29

**Authors:** Mikhail A. Kaplan, Artem D. Gorbenko, Alexander Y. Ivannikov, Bakhyt B. Kartabaeva, Sergey V. Konushkin, Konstantin Y. Demin, Alexander S. Baikin, Konstantin V. Sergienko, Elena O. Nasakina, Igor O. Bannykh, Irina V. Gorudko, Alexey G. Kolmakov, Alexander V. Simakin, Sergey V. Gudkov, Alexey P. Glinushkin, Mikhail A. Sevostyanov

**Affiliations:** 1A.A. Baikov Institute of Metallurgy and Materials Science, Russian Academy of Sciences (IMET RAS), 119334 Moscow, Russia; 2All-Russian Research Institute of Phytopathology, (VNIIF), 143050 Moscow, Russia; 3Department of Biophysics, Belarusian State University, 220006 Minsk, Belarus; 4Prokhorov General Physics Institute of the Russian Academy of Sciences, 119991 Moscow, Russia

**Keywords:** corrosion-resistant steel, silver, plasma dispersion, spherical powder, antibacterial properties, microstructure, chemical composition

## Abstract

The article is devoted to the study of melted ingots, plates rolled from them, and the resulting spherical powder made of corrosion-resistant 316L steel with the addition of 0.2 wt.% and 0.5 wt.% Ag. The study of antibacterial properties, microstructure, and distribution of silver concentrations, as well as qualitative analysis of silver content was carried out. The optimal mode of homogenization annealing of the ingot was 1050 °C for 9 h, which leads to the formation of an austenitic structure. It is shown that the addition of a small amount of silver does not affect the formation of the austenitic structure and silver is distributed evenly throughout the volume of the ingot. The austenitic structure also prevails in the plates after rolling. Silver is distributed evenly throughout the entire volume of the plate. It is noted that the addition of 0.2 wt.% Ag does not affect the strength, elongation, and microhardness of steel, and the addition of 0.5 wt.% Ag does not significantly reduce the strength of steel, however, all samples meet the mechanical characteristics according to the ASTM A240 standard. The qualitative chemical composition of samples made of corrosion-resistant steels was confirmed by X-ray fluorescence analysis methods. By the method of energy-dispersion analysis, the presence of a uniform distribution of silver over the entire volume of the powder particle was determined. The particles have a spherical shape with a minimum number of defects. The study of the antibacterial activity of plates and powder shows the presence of a clear antibacterial effect (bacteria of the genus Xanthomonas campestris, Erwinia carotovora, Pseudomonas marginalis, Clavibacter michiganensis) in samples No. 2 and No. 3 with the addition of 0.2 wt.% and 0.5 wt.% Ag.

## 1. Introduction

Corrosion-resistant steels have found wide application in various industries due to their resistance to general corrosion, high temperature oxidation, durability, high strength, and ductility [1,2,3,4]. One of the most popular steels is corrosion-resistant austenitic steel 316L, used in the manufacture of many products where maximum corrosion protection is required [5,6,7,8,9].

The use of traditional grades of austenitic steels in the manufacture of products in contact with aggressive media can contribute to protein adsorption, biofilm formation, and, as a consequence, corrosion or the formation of a source of bacterial infection [10,11,12,13]. To eliminate this problem, the material must have antibacterial properties. Studies of scientists have shown that modifying corrosion-resistant steels by adding silver to the composition can give them antibacterial properties [14,15,16,17,18,19,20].

In the study [14], 0.1, 0.2, and 0.3 wt.% Ag were added to corrosion-resistant 304 steel, and were smelted in an induction furnace in nitrogen medium to determine the effect of silver on antimicrobial and antibacterial activity. The results show that the content of 0.2 and 0.3 wt.% Ag gives the alloy 99.5% and 99.9% antibacterial activity, respectively, to the bacteria Staphylococcus aureus and Escherichia coli. In the study [15], 0.03 and 0.09 wt.% Ag were added to 316 steel, which also show an increase in antibacterial activity to the bacterium Escherichia coli with an increase in the Ag content [15]. In the study [16], the effect of adding 0.2 wt.% Ag in duplex stainless steel 2205 was studied. It is shown that the antibacterial properties of the material are 100% resistant to Escherichia coli and 99.5% resistant to Staphylococcus aureus. These studies show the feasibility of adding a small amount of silver as an alloying element to corrosion-resistant steel.

For further production of products made of corrosion-resistant steel, different methods of obtaining blanks can be used. Casting, forging, stamping, precise cutting, pressing, cutting, manufacturing from rolled products, welding, or a combined method are often used. Methods for obtaining blanks are selected taking into account the requirements: accuracy, the nature of the raw material base (homogeneous or heterogeneous structures can be combined), cost, technical characteristics, physico-chemical properties, and other parameters. The methodology is selected using profitability analysis and complex calculations.

There is also an active introduction of austenitic steels into additive technologies in the manufacture of various products. Additive methods are in demand due to the need for rapid production of products of complex geometric shape [21,22,23,24,25,26,27]. The raw material for additive methods is spherical powder or wire [28,29,30,31,32,33].

Therefore, obtaining and researching various blanks from new corrosive steel for the further manufacture of products with antibacterial properties is an urgent task.

The purpose of this work was to analyze the antibacterial properties, microstructure, and distribution of silver concentrations by volume of the material, as well as qualitative analysis of silver in ingots, plates rolled from it, and spherical powders obtained from corrosion-resistant 316L steel with the addition of 0.2 wt.% and 0.5 wt.% Ag.

## 2. Materials and Methods

Three compositions of corrosion-resistant 316L steel were smelted (sample No. 1 is the initial steel). Next, 0.2 wt.% Ag was added to sample No. 2, and 0.5 wt.% Ag was added to sample No. 3. The detailed chemical composition is presented in Table 1.

The melting of the canopies was carried out in an argon arc furnace with a non-consumable tungsten electrode LK200DI from Leybold-Heraeus (Leybold-Heraeus, Cologne, Germany). The ingots were subjected to homogenizing annealing in a vacuum of 2 × 10^−5^ mmHg at a temperature from 900 to 1050 °C for 9 h in a vacuum furnace ESQVE-1, 7.2, 5/21 SHM13 ((LLC "Scientific Production Enterprise" NITTIN ", Belgorod, Russia). A detailed technology for obtaining ingots is presented in [34].

For further studies of the effect of the addition of silver and titanium on the properties of the samples, plates with a thickness of 1 ± 0.1 mm were obtained by rolling.

The deformation of cast blanks was carried out by hot rolling on a double-roll mill DUO-300 (CJSC Istok ML, Nizhny Novgorod, Russia), with partial absolute compression per pass: 2 mm to the thickness of the workpiece 4 mm (13–20% per pass), then 1.0 mm to the thickness of the workpiece 2.0 mm (6–10% per pass), then 0.5 mm to the final thickness of the workpiece 1 ± 0.1 mm (3–5% per pass). The workpieces were heated before each deformation in a KYLS 20.18.40/10 muffle furnace by HANS BEIMLER (HANS BEIMLER, Berlin, Germany) for 20–25 min to a temperature of 1100 °C before the first rolling and for 5 min during intermediate heating. After heating, the workpiece was rolled with cold rolls, which contributed to a sharp cooling.

Spherical powder was also obtained from the melted ingots by rolling, rotary forging, drawing, and further plasma spraying of wire. The detailed technology is presented in [34].

To study the structure, the slots were obtained by pressing samples on a pneumohydraulic press IPA 40 (Remet, Milan, Italy) at a temperature of 170 °C and exposure for 20 min into Aka-Resin epoxy resin. The preparation of samples for metallographic examination was carried out at the Buehler Phoenix 4000 installation (Buehler, Lake Bluff, Illinois, USA) by sequential grinding and polishing after pressing on a Piatto diamond disc with a grain size of P120 for 3 min, P320 for 5 min, P600 for 5 min; on a diamond disc of fine grinding Aka-Allegran-3 with DiaMaxx Poly suspension with diamond particle sizes of 6 microns for 5 min; and on Akasel NAPAL velvet with DiaMaxx Poly suspension with diamond particle sizes of 3 and 1 microns for 3–5 min on each. Etching of the surface of the samples was carried out with a mixture of acids for high-alloy steels, consisting of 20% nitric acid (HNO_3_), 10% sulfuric acid (H_2_SO_4_), 5% hydrofluoric acid (HF), and 65% water (H_2_O). The etching duration was from 10 to 20 min. After etching, the slates were washed with distilled water and ethyl alcohol.

Optical microscopy was performed on an Altami MET 5S microscope (Altami, St. Petersburg, Russia) using a video camera with a resolution of 14 megapixels built into the device and special Altami Studio 4.0 software.

Morphology, microstructure, and mapping (determination of the concentration distribution of chemical elements) for ingots and powder were studied using a scanning electron microscope JEOL JSM-IT500 (JEOL, Tokyo, Japan) with a power of 15 kW with the prefix of energy-dispersive microanalysis INCA ENERGY. For the rolled plates, an electron microscope Tescan Vega II SBU (TESCAN, Brno, Czech Republic), Tescan company with equipment for energy-dispersive microanalysis (INCA Energy 300, Oxford Instruments company (Oxford Instruments, Abingdon, UK)), was used. During the study, images of the sample surface with high spatial resolution obtained in the secondary electron mode were analyzed. Mapping (distribution of chemical composition) was also carried out. The slates of all compositions were prepared as samples for the study. An electrically conductive resin was used in the manufacture of the grinds.

Qualitative analysis of the elemental composition was performed on an X-ray fluorescence wave dispersion spectrometer of the BRUKER S8 Tiger sequential type (BRUKER, Karlsruhe, Germany). For X-ray fluorescence analysis, slates of all compositions were prepared as samples for the study.

To determine the antibacterial activity of the samples, bacteria of the genus Xanthomonas campestris, Erwinia carotovora, Pseudomonas marginalis, and Clavibacter michiganensis were used. Colonies of bacteria grown in a test tube were transferred to a test tube with sterile distilled water to create a bacterial suspension. Then samples were placed on the sowing surface and kept for 5 days at a temperature of 28 °C. The antibacterial activity for sowing was evaluated by the phenomenon of bacterial growth retardation around the material.

## 3. Results

Micrographs of the ingot structure of samples No. 1 (316L), No. 2 (+0.2 wt.% Ag), and No. 3 (+0.5 wt.% Ag) after smelting and homogenization annealing at 900 °C, 950 °C, 1000 °C, and 1050 °C are shown in Figure 1.

After smelting, uneven structure and predominance of dendritic structure are observed on all ingots, which may indicate liquation (heterogeneity of chemical composition). After homogenization annealing at temperatures of 900 °C and 950 °C, the dendritic structure is preserved for 9 h. During homogenization annealing at a temperature of 1000 °C, the dendritic structure is partially preserved for 9 h, but partial recrystallization of the alloy is observed. For complete recrystallization, homogenization annealing is used for 9 h at a temperature of 1050 °C, which leads to the alignment of the structure and the formation of equiaxed austenite grains with a size of ~50 microns. A further increase in temperature is impractical, due to an increase in grain, which can lead to difficulties with further plastic deformation.

Also, surface images were obtained using a scanning electron microscope and spectral analysis (mapping) was carried out (Figure 2).

As can be seen from Figure 2, the mapping shows a uniform distribution of silver in ingots of compositions No. 2 and No. 3.

The rental was carried out with preheating to a temperature of 1100 °C. After heating, the ingot was rolled, which contributed to a sharp cooling. The structures and distribution of silver over the area of the rolled plates of samples No. 1, No. 2, and No. 3 obtained with an optical microscope and a scanning electron microscope are shown in Figure 3.

A grain structure is observed in the plates, which may indicate that recrystallization occurs after the plate exits the rolls. A grain structure is also observed on a scanning electron microscope. Mapping shows a uniform distribution of silver over the area of the plates in compositions No. 2 and No. 3.

The morphology of powder particles obtained by flame atomization was studied (Figure 4). The morphology of the powder particles with the addition of silver does not differ from the initial composition. The powder obtained by plasma atomization shows high sphericity and roundness with a minimum number of defects. Mapping shows a uniform distribution of silver on the surface of powder particles of compositions No. 2 and No. 3.

Also, images of the surface of the powder particles in secondary electrons (SED) and topographic and phase contrast (BED-C) in the back-reflected ones were obtained using a scanning electron microscope, and spectral analysis of the surface (mapping) was carried out (Figure 5).

As can be seen from Figure 5, the particles in the cross section have a spherical shape and a grain structure. Mapping shows a uniform distribution of silver in powder particles of compositions No. 2 and No. 3. This distribution should improve the antibacterial properties of the samples and can increase resistance to pitting and inter-crystalline corrosion.

Also, for qualitative analysis, X-ray fluorescence analysis was performed, which shows the presence of silver in the samples, which indicates that all blanks contain silver. This method is an express method that does not determine the exact chemical composition. A qualitative analysis was carried out in [34] on the Analytik Jena PlasmaQuant 9100 optical emission spectrometer (Analytik Jena, Jena, Germany). The results show that the silver content does not change during the production of blanks of various form factors and corresponds to the content in composition No. 2—0.179 ± 0.073% and in composition No. 3—0.4972 ± 0.171% for ingot and 0.1956 ± 0.075%, 0.4851 ± 0.178% for powder, respectively [34]. The examination of the plates for the silver content also shows values for composition No. 2—0.1983 ± 0.062% and for composition No. 3—0.4964 ± 0.163%.

The study of antibacterial activity shows the presence of a clear antibacterial effect in samples No. 2 and No. 3 with the addition of 0.2 wt.% and 0.5 wt.% Ag. As a result of the experiment conducted in Petri dishes with samples No. 2 and No. 3, one can see a clear antibacterial effect around the samples in comparison with the control (a sterile zone around the studied samples) and, thereby, suppression of bacterial growth and development. Figure 6 and Figure 7 show a decrease in the number of bacteria due to the suppression of bacterial growth and development. Composition No. 2, with the addition of 0.2 wt.% Ag, shows an antibacterial effect to bacteria of the genus Pseudomonas marginalis and Clavibacter michiganensis, and composition No. 3, with the addition of 0.5 wt.% Ag, shows an antibacterial effect to bacteria of the genus Xanthomonas campestris, Erwinia carotovora, and Clavibacter michiganensis (Table 2).

The obtained results of the evaluation of the effect of silver on antibacterial properties show that an alloy with a low concentration of silver 0.2 wt.% provides an antibacterial effect to the bacteria Pseudomonas marginalis and Clavibacter michiganensis, but does not show a pronounced antibacterial effect to the bacteria Xanthomonas campestris, or Erwinia carotovora. An increase in the concentration of silver in the alloy provides an antibacterial effect to bacteria of the genus Xanthomonas campestris, Erwinia carotovora, and Clavibacter michiganensis, but does not show an effect for bacteria of the genus Pseudomonas marginalis. These patterns are manifested both for plates and for spherical powders. Therefore, with the possible production of products by additive layer-by-layer methods, a gradient structure can be formed using spherical powders with different concentrations of silver, which should provide a synergistic effect and prevent bacterial contamination of the surface from all the bacteria studied. The proposed method of using spherical powders can be implemented by directed energy deposition (DED) methods [35,36]. In practice, with additive manufacturing, spherical powders of different chemical compositions should be filled into separate bunkers.

## 4. Conclusions

Ingots, rolled plates, and spherical powders made of corrosion-resistant steel with the addition of 0.2 wt.% Ag and 0.5 wt.% Ag are obtained in the ingots after smelting, and the dendritic structure prevails. The optimal regime of homogenization annealing (1050 °C for 9 h) is revealed, which leads to the formation of an austenitic structure, its alignment, and the formation of equiaxed grains with a size of ~50 microns. It is shown that the addition of a small amount of silver does not affect the formation of the austenitic structure and silver is distributed evenly throughout the volume of the ingot.

According to the results of optical and scanning microscopy, it should be said that an austenitic structure is observed in the plates. Silver is distributed evenly throughout the entire volume of the plate.

The qualitative chemical composition of samples made of corrosion-resistant steels is confirmed by X-ray fluorescence analysis methods.

By the method of energy-dispersion analysis, the presence of a uniform distribution of silver over the entire volume of the powder particle is determined. The particles have a spherical shape with a minimum number of defects.

The study of the antibacterial activity of plates and powder shows the presence of a clear antibacterial effect in samples No. 2 and No. 3 with the addition of 0.2 wt.% and 0.5 wt.% Ag. Composition No. 2, with the addition of 0.2 wt.% Ag, shows an antibacterial effect to bacteria of the genus Pseudomonas marginalis and Clavibacter michiganensis, and composition No. 3, with the addition of 0.5 wt.% Ag, shows an antibacterial effect to bacteria of the genus Xanthomonas campestris, Erwinia carotovora, and Clavibacter michiganensis.

## Figures and Tables

**Figure 1 materials-16-00319-f001:**
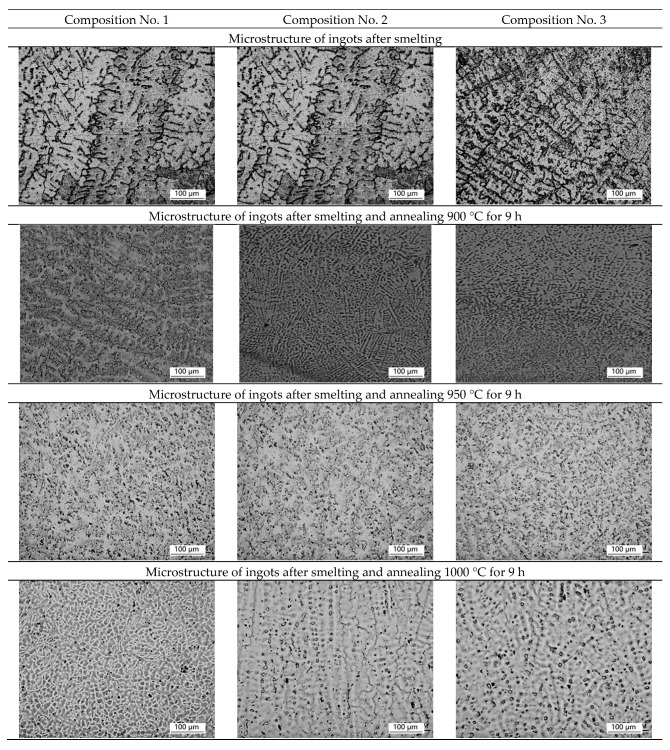
The effect of the method of production on the microstructure of ingots (optical microscope).

**Figure 2 materials-16-00319-f002:**
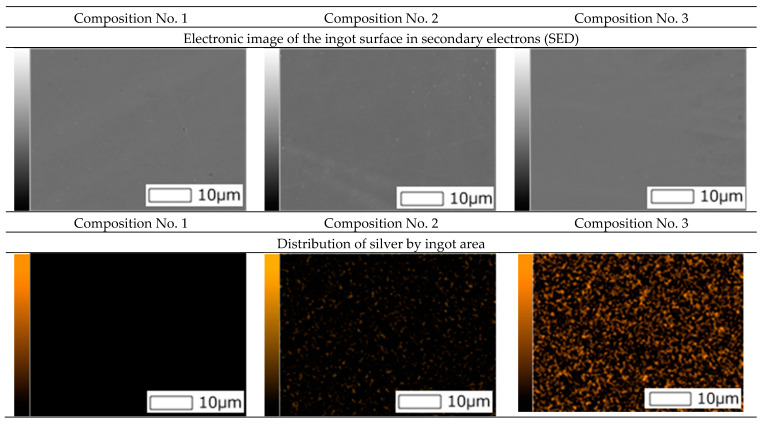
Electronic image of the ingot surface in secondary electrons and distribution of silver according to EDS data.

**Figure 3 materials-16-00319-f003:**
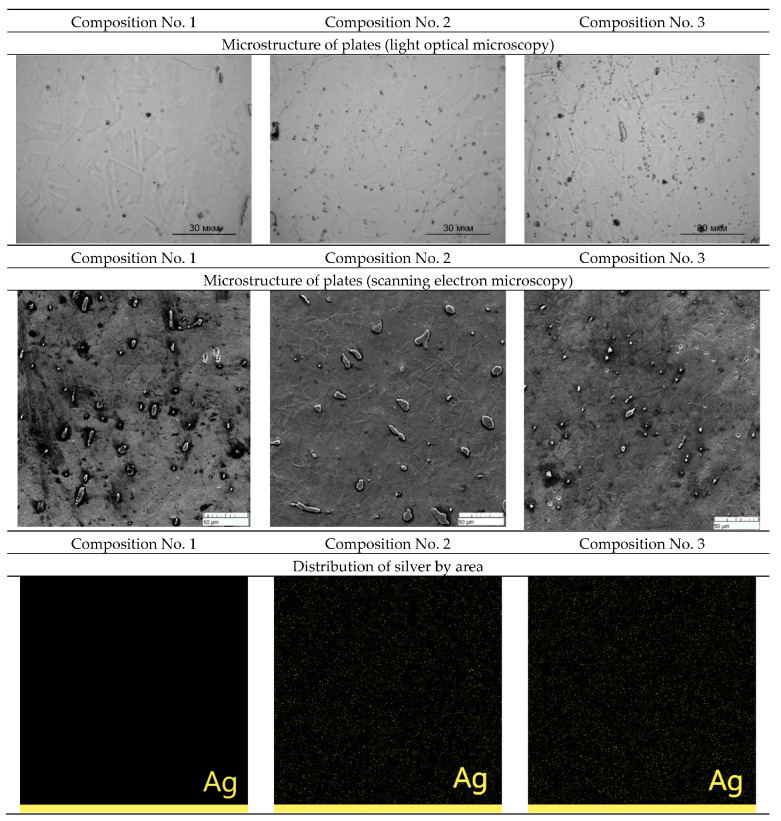
Influence of silver content on the microstructure of steel plates.

**Figure 4 materials-16-00319-f004:**
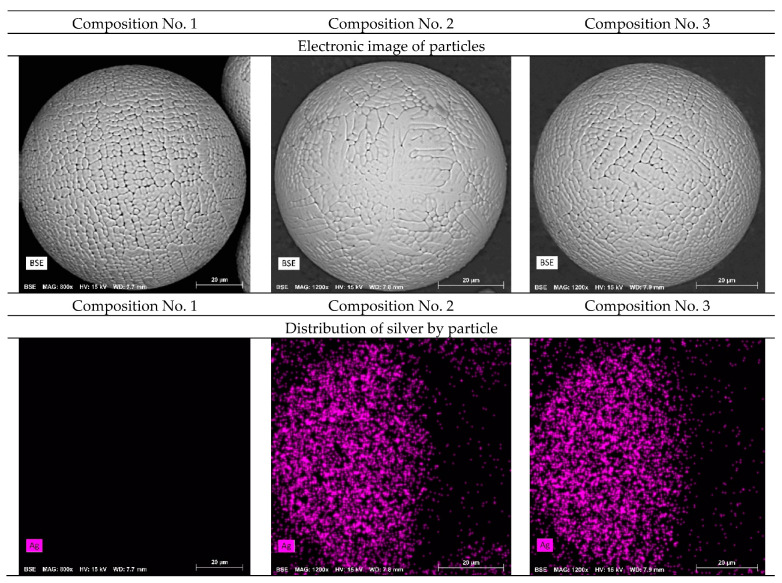
SEM images of powder particles and the distribution of silver on the surface (No. 1—316L; No. 2—316L +0.2 wt.% Ag; No. 3—316L +0.5 wt.% Ag).

**Figure 5 materials-16-00319-f005:**
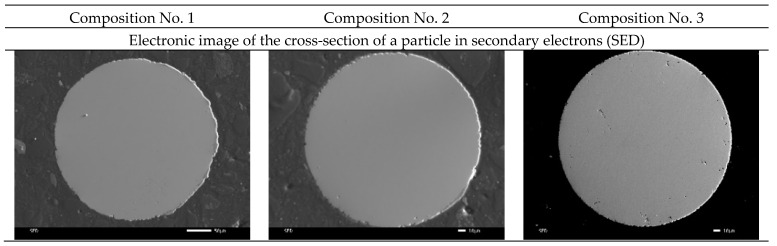
SEM images of the cross-section of powder particles and the distribution of silver over the area (No. 1—316L; No. 2—316L +0.2 wt.% Ag; No. 3—316L +0.5 wt.% Ag).

**Figure 6 materials-16-00319-f006:**
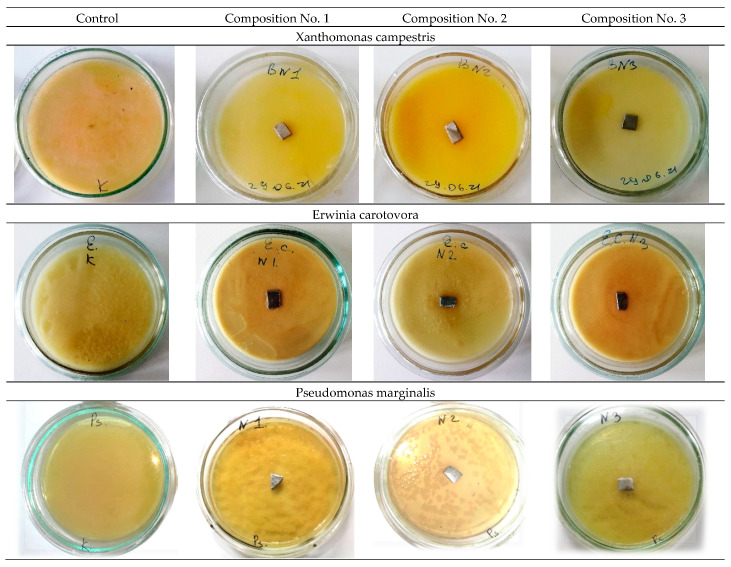
Samples of plates in bacterial suspensions (No. 1—316L; No. 2—316L +0.2 wt.% Ag; No. 3—316L +0.5 wt.% Ag).

**Figure 7 materials-16-00319-f007:**
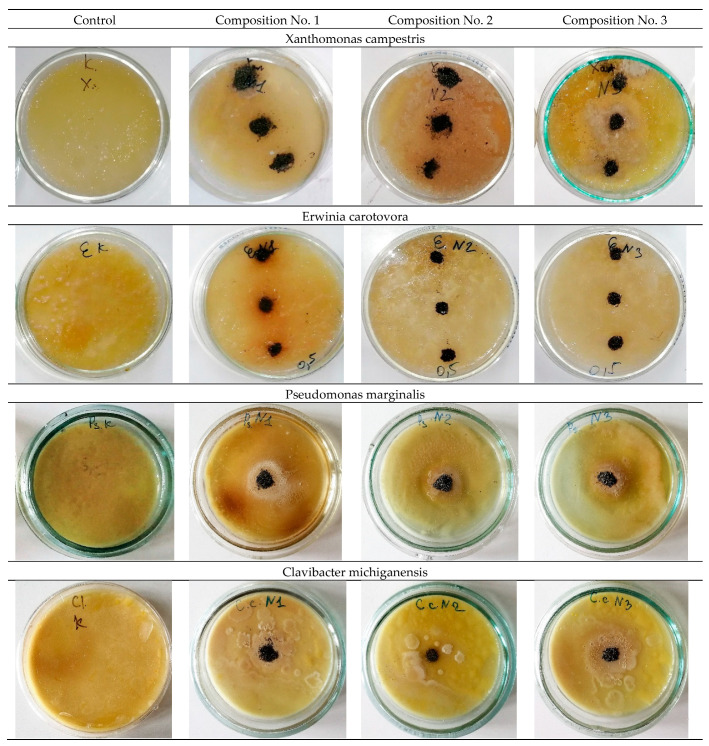
Powder samples in bacterial suspensions (No. 1—316L; No. 2—316L +0.2 wt.% Ag; No. 3—316L +0.5 wt.% Ag).

**Table 1 materials-16-00319-t001:** Chemical composition of steels (wt.%).

Steel	C,%	Cr,%	Ni,%	Ag,%	Si,%	Mn,%	Mo,%	Si,%
No. 1	0.023	17	10	0	0.5	1.5	2	0.5
No. 2	0.023	17	10	0.2	0.5	1.5	2	0.5
No. 3	0.023	17	10	0.5	0.5	1.5	2	0.5

**Table 2 materials-16-00319-t002:** Diameter of the sterile zone of the samples.

Variant	Form	Type of Bacteria			
		Xanthomonas Campestris	Erwinia Carotovora	Pseudomonas Marginalis	Clavibacter Michiganensis
		Diameter of the Sterile Zone (cm)			
No. 1	Plate	-	-	-	-
No. 1	Powder	-	-	-	-
No. 2	Plate	-	-	1.5	1.6
No. 2	Powder	-	-	1.3	-
No. 3	Plate	1.8	1.7	-	1.8
No. 3	Powder	1.7	1.7	-	1.7

## Data Availability

Not applicable.

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
