# Peer review of "Investigation of Antibacterial Properties of Corrosion-Resistant 316L Steel Alloyed with 0.2 wt.% and 0.5 wt.% Ag"

_materials, 2022, doi:10.3390/ma16010319_

Round 1

Reviewer 1 Report

Comment:

This manuscript reports on Ag-doped SS316 for enhanced antibacterial properties. The authors synthesize the samples with different Ag composition and measure the bacteria growth with the metal powder. However, there are remaining questions on the elemental ratio and control group. Thus, I think a minor revision is needed for this manuscript before it published on Materials at this stage.

Specific comments:

1.The Ag doping of 0.2% and 0.5% is only determined by the initial precursor ratio. Quantitative analysis of the composition is necessary for the confirmation of the Ag doping.

2. The authors provide the surface elemental mapping and confirm the existence of Ag. What is the surface Ag concentration? Is it different from the bulk composition?

3. For the antibacterial properties measurement, any explanation of the different results on different bacteria for 0.2% Ag and 0.5% Ag samples? Why some favors 0.2% Ag and other favors 0.5% Ag?

Author Response

Thank you for your useful comments and suggestions on the structure of our manuscript. Please see the attachment, there is the text of the corrected article.

  1. We agree with the comment. Necessary corrections were made. Added quantitative composition analysis.
  1. We agree with the comment. Necessary corrections were made. The EDS analysis of the distribution of silver in the cross section of a spherical powder qualitatively evaluates its uniform distribution. EDS analysis is semi-quantitative. To estimate the silver content in the cross section and on the surface, an EDS pattern of silver distribution on the surface of a spherical particle is presented. Quantitative analysis indicates the uniform distribution of silver on the surface and over the cross section of the particle.
  1. For 3 species of bacteria, 0.2% Ag was not enough. The effect occurred only at 0.5% Ag. However, this did not work for Pseudomonas marginalis, and this effect will be studied further.

Reviewer 2 Report

The study presented in this research is sound, and the results produced are interesting. But a revision is required, and after responding to the following remarks and revising the paper, the manuscript may be considered for publication.

1. Literature review needs to include several recent, relevant publications (high impact) highlighting their key findings. The current version only discussed general aspects while the review of each from several papers is necessary. You may provide a review summary table consisting of a column for the comments or key conclusions.

2. More recent relevant literature or similar work discussion is mandatory in the introduction section, which is missing in the Introduction. Authors are suggested to add one paragraph in the introduction section by discussing the recent progress and citing similar work.

3. The novelty of the work is missing in the introduction. Authors are suggested to include a separate paragraph discussing the novelty and importance of the present work.

4. Authors are suggested to include a literature review on the recent publication on the following references in the introduction section: DOIs: 10.1016/j.ceramint.2022.07.220; 10.1039/d2ra05304g; 

5. Reduce the similarity. Check attached report.

6. Also, check the typos throughout the manuscript during revision submission.

Author Response

Thank you for your useful comments and suggestions on the structure of our manuscript. Please see the attachment, there is the text of the corrected article.

    1. We agree with the comment. Necessary corrections were made. We have made the necessary changes to the introduction.
    2. We agree with the comment. Necessary corrections were made. We have made the necessary changes to the introduction.
    3. We agree with the comment. Necessary corrections were made. We have made the necessary changes to the introduction.
    4. Thanks for the interesting article. These works have been added to the analysis. The introduction text has been modified.
    5. The results, discussion and conclusions are original. The similarity in the Materials and Methods section has been reduced.
    6. We agree with the comment. Necessary corrections were made. The text of the article has been technically edited

Reviewer 3 Report

Through the review, I think this is a very nicely written paper, and the author analyzed and discussed the results in detail and correctly. The analysis developed in this paper is correct and the obtained results are interesting. The paper has sufficient novelty which covers the scope of the journal. Therefore, the manuscript may consider for publication in the Materials journal after responding to the following comments and revising the manuscript properly.

- Provide a more appealing title with no acronyms in a precise and concise manner.

- Omit trivial information.

- Explain in brief how the present paper differs from the published ones.

- State-specific objectives.

- Get its English edited very carefully.

- Provide better quality figures.

- State the main findings in the conclusions.

Author Response

Thank you for your useful comments and suggestions on the structure of our manuscript. Please see the attachment, there is the text of the corrected article.

  1. The presented title corresponds to the topic and is consistent with the headings of articles in the field of research on the antibacterial properties of corrosion-resistant steels.
  2. We agree with the comment. Necessary corrections were made.
  3. This article differs in that this brand did not add this amount of silver, did not receive powder and plates, and did not study antibacterial properties on such bacteria.
  4. We agree with the comment. Necessary corrections were made.
  5. We agree with the comment. Necessary corrections were made.
  6. We agree with the comment. Necessary corrections were made.
  7. We agree with the comment. Necessary corrections were made.
